
# The Role of Unmanned Aerial Vehicles (UAVs) In Monitoring Rapidly Occuring Landslides

**Servet Yaprak[1], Omer Yildirim[1], Tekin Susam[1], Samed Inyurt[2] Irfan Oguz[1]**

[1]*Geomatics Engineering, Gaziosmanpaşa University, Tokat, Turkey*
[2]*Geomatics Engineering, Bulent Ecevit University, Zonguldak Turkey*
*Corresponding author: Samed INYURT, e-mail:samed_inyurt@hotmail.com*

**Abstract:** This study used the Unmanned Aerial Vehicle (UAV), which was designed and produced to monitor rapidly occurring landslides in forest areas. It was aimed to determine the location data for the study area using image sensors integrated into the UAV. The study area was determined as the landslide sites located in the Taşlıçiftlik Campus of Gaziosmanpaşa University, Turkey. It was determined that landslide activities were on going in the determined study area and data was collected regarding the displacement of materials. Additionally, it was observed that data about landslides may be collected in a fast and sensitive way using UAVs, and this method is proposed as a new approach. Flights took place over a total of five different periods. In order to determine the direction and coordinate variables for the developed model, eight Ground Control Points (GCPs), whose coordinates were obtained with the GNSS method, were placed on the study area. In each period, approximately 190 photographs were investigated. The photos obtained were analysed using the PIX4D software. At the end of each period, the RMS and Ground Sample Distance (GSD) values of the GCPs were calculated. Orthomosaic and Digital Surface Models (DSM) were produced for the location and height model. The results showed that max RMS=±3.3 cm and max GSD=3.57cm/1.40 in. When the first and fifth periods are compared; the highest spatial displacement value ΔS = 111.0 cm, the highest subsidence value Δh = 37.3 cm and the highest swelling value Δh = 28.6 cm as measured.

**Keywords:** Unmanned Aerial Vehicles (UAV); landslides; ground sample distance (GSD); digital surface model (DSM); orthomosaic

## 1. Introduction

Landslides are a worldwide phenomenon that create dramatic physical and economic effects and sometimes lead to tragic deaths. During landslides two main factors occur, which are human and environmental effects. The human factors may be controlled; however, it is very difficult to control the topography and soil structure (Turner et al., 2015). Thus, landslides cause disasters on a global scale each year. These disasters are increasing in number due to the incorrect usage of the land. The main reason for the increase in landslide disasters is the instability of the soil and erodibility on the surface. Surface soil erodibility takes place as a result of various issues, such as deforestation, an increase in consumption by an increasingly larger population, uncontrolled land usage, etc. (Nadim et al., 2006). Landslides are primarily disasters that take place in mountainous and sloped areas around the world (Dikau et al., 1996). Landslides do not always show characteristic occurrences, however, they are usually triggered by increased stress on sloped surfaces. This triggering can occur faster because of short or long periods of heavy rain, earthquakes, or subterranean activity (Lucier et al., 2014). During landslide monitoring, a number of factors need to be continuously assessed, including the: extent





of the landslide, detection of fissure structures, topography of the land and rate of
displacements that could be related to fracture (Niethammer et al., 2010). Understanding
the mechanism of landslides may be made easier by being able to measure the vertical
and horizontal displacements. This is possible by forming a Digital Surface Model (DSM)
of the landslide area.
The calculation of displacements by Differential GPS (DGPS), total station, airborne
Light Detection and Ranging (LIDAR) and Terrestrial Laser Scanner (TLS) techniques
have been used since the beginning of the 2000s (Nadim et al., 2006). Additionally,
remote sensing has been put into operation in combination with other techniques
(Mantovani et al., 1996). There are several platforms, which are used to monitor landslide
occurrences via the method of remote sensing, where displacement data can be collected.
These include remote sensing satellites, manned aerial vehicles, specially equipped land
vehicles and, as a new method, Unmanned Aerial Vehicles (UAV) (Rau et al., 2011).
These UAV are aerial vehicles that are able to fly without crew automatically or semi-
automatically based on aerodynamics principles. UAV systems have become popular in
solving problems in various fields and applications (Saripalli et al., 2003; Tahar et al.,
2011). In parallel with the developing technology, UAVs have been used in recent years
in integration with the Global Positioning System (GPS), Inertial Measurement Units
(IMU) and high definition cameras and they have also been used in remote sensing (RS),
digital mapping and photogrammetry in scientific studies. While satellites and manned
aerial vehicles are able to gather location data in high resolutions of 20-50 cm/pixel,
UAVs are able to obtain even higher resolutions of 1 cm/pixel, as they are able to fly at
lower altitudes (Hunt et al., 2010). Indeed, UAV Photogrammetry opens up various new
applications in close-range photogrammetry in the geomatics field (Eisenbeiss 2009).
Monitoring landslides using UAV systems is an integrated process involving ground
surveying methods and aerial mapping methods. All measurement devices that require
details are integrated to UAVs, which fly at lower altitudes than satellites or planes. All
positional data are collected safely from above, except for determining and measuring the
control points (Nagai et al., 2008).
This study was conducted in the landslide site at the Organized Industrial Zone near a
campus of Gaziosmanpaşa University. The area of the studied field was approximately
50 hectares. The Multicopter was produced by the Department of Geomatics Engineering
at Gaziosmanpaşa University (GOP) and the firm TEKNOMER was used for this study.
A Sony Alpha 6000 (Ilce 6000) camera, IMU and GPS systems, produced for moving
platforms, were integrated to the UAV. Five different flights took place on different dates
in the study area and an average of 290 photographs were obtained on each flight. Eight
ground control points (GCPs), which were well distributed over the data area, were set
up in the landslide area (Figure 6). The positional information about the ground control
points was collected using four dual-frequency Geodesic GNSS receivers (Trimble,
Topcon). Two hours of static GNSS measurements were analyzed in 3D using the Leica
LGO V.8.3 software in connection to the TUSAGA Active System.

**2. System Design**





This study used the multicopter, which was produced by the department of
Geomatics Engineering at Gaziosmanpaşa University (GOP) (Figure 1a and b). The
designed multicopter consisted of a platform and camera systems.

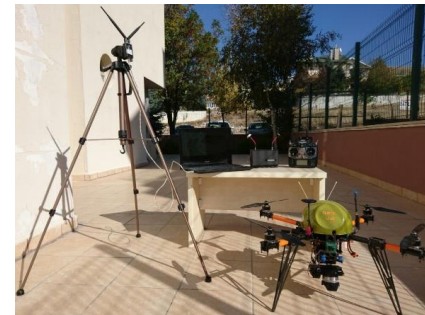 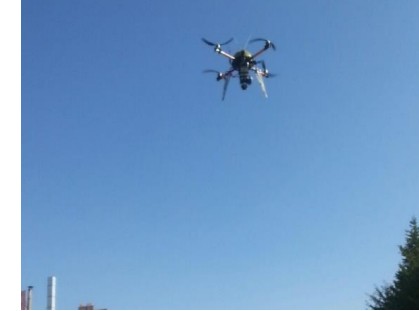

Figure 1a. The UAV and environmental components          Figure 1b. The UAV in the air
**2.1. UAV Platform**
UAV platforms provide crucial alternative solutions for environmental research
(Nex and Remondino, 2014). The UAV environmental components used in this study
were integrated into the multicopter as seen Figure 2. The platform had a blade-span of
0.80 m, height of 0.36 m, weight of 4.4 kg and operating weight of 5 kg. All sensors were
placed on the carrying platform to achieve operating integrity. The carrying platform
operated at the speed of 14 m/sec while shooting photos. The multicopter had a stabilized
camera gimbal to take nadir photos during the flight. The characteristics of the carrying
platform are given in Table 1.

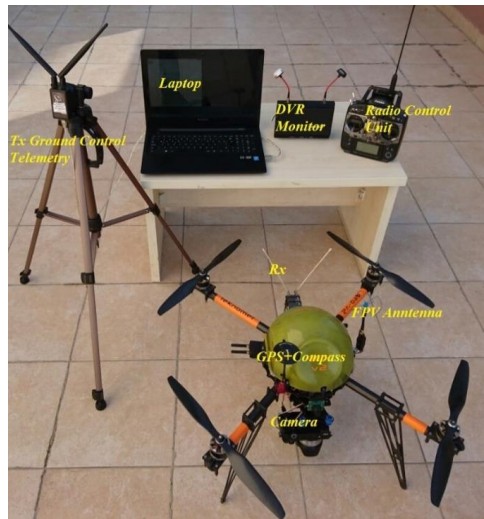


Figure 2. UAV environmental components


Table 1. Platform technical specifications

| Specification | Technical Details |
| --- | --- |
| Weight | 4.3 kg |
| Wing Span | 74 cm |
| Payload | 4 kg |
| Height | 34 cm with GPS Antenna |
| Range | 4 km |
| Endurance | 30 min |
| Speed | 14 m/sec |
| Maximum Speed | 70 km - 30 mm /sec |
| Radio Control | 433 MHz |
| Frame Transponder (FPV) | 2.4 GHz |
| Telemetry Radio | 868 MHz |
| GPS | 5 Hz – 72 channels |
| Battery | 6S li-po 25C 1600 Mah |
| Monitor | 40 Channels 5.8 GHz DVR 7 inch LED system |
| Gimbal | Mapping Gimbal |
| Motors | 35 x 15 Brushless Motor |
| Frame | 22 mm 3K Carbon |
| ESC | 60 Ampere 400 Hz |
| Prop | 15 x 55 inch Carbon |


### 2.2. Camera System
In this study, a Sony ILCE-6000 E16mm F2.8-16.0-6000x4000 (RGB) camera
was used for collecting visible imagery (Figure 3). Table 2 shows the characteristics of
the camera. The main controller of the UAV was programmed to shoot photos regularly,
every two seconds. This way, the shutter of the camera was triggered at the desired
frequency intervals.
The camera and the main flight controller card were connected using a special
cable. Vibration isolation materials were used between the camera and the UAV to
prevent the effects of flight vibrations on the camera. During the flight, all photos were
taken in the RAW format and stored in the memory of the camera.



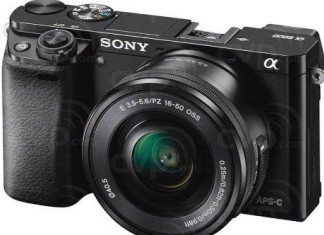



Figure 3. The camera used in the study

Table 2. Technical properties of the camera
(http://pdf.crse.com/manuals/4532055411.pdf[Accessed 2017 May 10)





| Property | Technical Detail |
|---|---|
| Dimensions | 4.72 x 2.63 x 1.78 in |
| Weight | 10.05 oz (Body Only) / 12.13 oz (with battery and media) |
| Megapixels | 12 MP |
| Sensor Type | APS-C |
| Sensor Size | APS-C type (23.5 x 15.6 mm) |
| Number of pixels (effective) | 24.3 MP |
| Number of pixels (total) | Approx. 24.7 megapixels |
| ISO sensitivity (recommended exposure index) | ISO 100-25600 |
| Clear image zoom | Approx. 2x |
| Digital zoom (still image) | L: Approx. 4x; M: Approx. 5.7x;S: Approx. 8x |
| LCD Size | 3.0 in wide type TFT LCD |
| LCD Dots | 921,600 dots |
| Viewfinder Type | 0.39 in-type electronic viewfinder (colour) |
| Shutter speed | Still images: 1/4000 to 30 sec, Bulb, Movies: 1/4000 to 1/4 (1/3 steps) up to 1/60 in AUTO mode (up to 1/30 in Auto slow shutter mode) |
| Flash sync. Speed | 1/160 sec. |


## 3. Study Area

This study was carried out in order to monitor the landslides with UAV in Tokat
Province. The study area was selected to track the landslides that began in the area where
factories and industrial enterprises are located. There is a great landslide risk in this
industrial area, it is a preexisting situation and if the motion continues or accelerates it
could mean great danger for the nearby factories. For this reason, the movement needs to
be monitored.

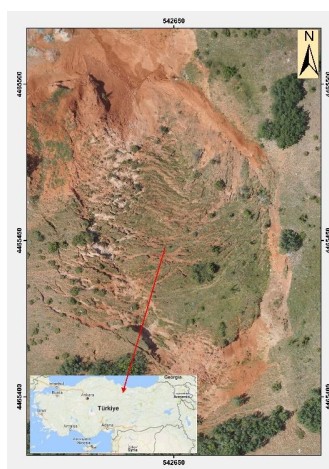






Figure 4. The study area

The coordinates of the landslide area used for the study are given as $40^0$ 19' 20.8"
N, $36^0$ 30' 0.6" E. The study area is shown in Figure 4.

### 3.1. Soil Properties of the Study Area

The oldest layer at the research area is Paleozoic aged metaophiolite (Metadunite,
amphibolite/Metagabbro). The sedimentary layer, which is called eosin aged "Çekerek
formation", is over the metaophiolite layer. This formation consists of sandstone, pebble,
silt and clay (Sumengen, 1998).

Soil samples were collected from three different locations at 0-0.2 and 0.2-0.4 m
depths and analyzed for soil particle distribution using the Bouyoucos hydrometer method
(Gee and Bauder, 1986). The fraction greater than 2 mm diameter was separated and
reported as coarse material (Gee and Bauder, 1986). The dispersion ratio was calculated
using Equation 1 (Middleton 1930). The aggregate stability index was calculated by the
wet sieving method (Yoder 1936).

Dispersion Ratio = {D (Silt + Clay) / T (Silt + Clay)} x 100         (1)

Where D is dispersed silt + clay after 1kg of oven-dried soil in a litre of distilled
water was shaken 20 times; T, is total silt + clay determined by the standard sedimentation
method in a non-dispersed state. Some soil properties of the study area are presented in
Table 3. The results of the mechanical analysis in most of the studied soils showed a high
clay and silt and low sand content. The textural classes of the soil objects were
determined as clay (C), clay loam (CL) and silt loam (SiL). The high clay and silt content
of study area increased disaggregation by leading to imbalances in the moisture content
of different soil layers instead of aggregation. This effect may result in high runoff, soil
loss and weathering processes. When the topsoil and subsoil layers are compared, the clay
content of the topsoil layer decreased, the silt content was the same and the sand content
increased at study site one. At study site two, the higher clay and lower silt contents were
detected more in the subsoil than in the topsoil. The same result was observed for study
site three. Textural differences between the topsoil and subsoil created moisture
differences in the soil layers and this situation may result in large mass movements. In
the study area, the coarse material varied between 4.2 and 31.0%, depending on the mass
transportation.
Table 3. Some soil properties of the study area

| Study Site | Soil Depth (m) | Texture | | | | Coarse Material % | Aggregate Stability % | Dispersion Ratio % |
|---|---|---|---|---|---|---|---|---|
| | | Clay % | Sand % | Silt % | Class | | | |
| 1 | 0.0-0.2 | 40.0 | 28.7 | 31.3 | CL | 13.0 | 34.3 | 36.9 |
| | 0.2-0.4 | 37.5 | 31.2 | 31.3 | CL | 31.0 | 41.3 | 60.0 |
| 2 | 0.0-0.2 | 50.0 | 11.2 | 38.8 | C | 4.2 | 13.9 | 57.8 |
| | 0.2-0.4 | 52.5 | 11.2 | 36.3 | C | 19.7 | 46.2 | 49.3 |
| 3 | 0.0-0.2 | 40.0 | 13.7 | 46.3 | SiL | 15.7 | 18.8 | 36.3 |
| | 0.2-0.4 | 42.5 | 13.7 | 43.8 | SiL | 6.6 | 13.1 | 47.9 |




To evaluate the forces on the soil resistance to the mass movement of the study
area, aggregate stability and dispersion ratio indexes were used. The aggregate stability
of the soil objects was under 46.2% and showed low aggregate stability with a high risk
of soil movement. The dispersion ratio index indicated a sharp boundary between erodible
and non-erodible soils, since a dispersion ratio greater than 10 indicated erodible soils
and less than 10 indicated non-erodible soils. The dispersion values of the study area were
greater than 10 with high erosion risk.

***3.2. 3D Ground Control Points***
A total of eight 3D GCPs were used in the study area. The GCPs were placed in a way so
that they could be easily seen in photos taken from above, near the landslide site, but
where future landslides would not affect them (Figure 5). All GCPs were placed as
concrete blocks, which were topped with side wings with dimensions of 40x15 cm so
they could be easily detected in the computer environment. The geometrical distribution
of the GCPs in the study area is given in Figure 6.

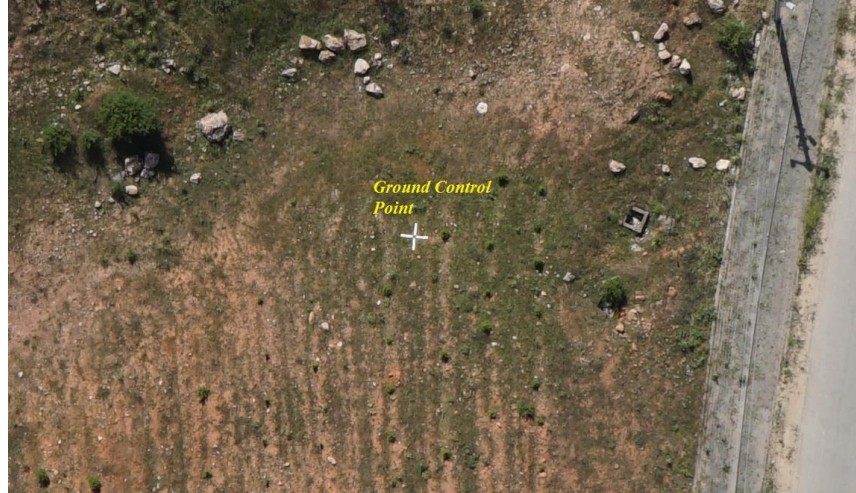

Figure 5. Ground Control Point (GCP)
The 3D positional information of the GCPs was collected by the CORS-TR
System (Mekik et al., 2011) using Topcon GR3 dual-frequency GNSS (Global
Navigation Satellite System) receivers. GNSS data was collected for a minimum of two
hours for each point and it was computed via static analysis at the datum of ITRF96 and
epoch of 2005.00. With the dual-frequency receivers used, the horizontal sensitivity of
the GCPs were found to be ±3mm+0.5 ppm, while the vertical sensitivity was found to
be ±5 mm+0.5 ppm.





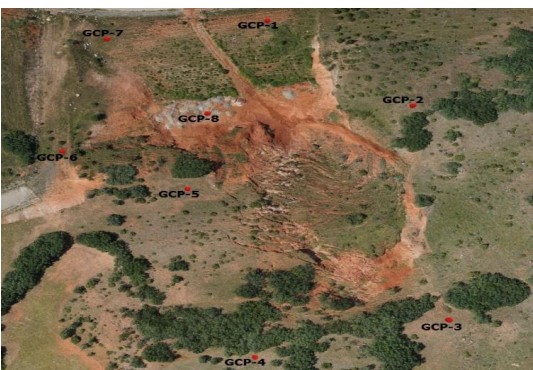


Figure 6. The geometric distribution of GCPs
### *3.3. Flight Planning and Shooting of the Photos*
Flight plans were made following the GNSS measurements of the GCPs and obtaining
their coordinates via analysis. The flights were carried out at five different periods
following rainfall or snowfall, where the landslide area was the most active. The flight
dates and flight altitude information are given in Table 4. The flight plan for the study
area was set within the Mission Planner software with vertical overlapping of 80%,
horizontal overlapping of 65%, a flight altitude of 100 meters and flying speed of 14
m/sec. A number of overlapping images were computed for each pixel of the
orthomosaics. The green areas indicated an overlap of over five images for every pixel
(Figure 8) (http://ardupilot.org/planner/docs/common-history-of-ardupilot.html accessed
2017 June 3. 2017). The prepared flight plan (Figure 7a, b) was uploaded onto the UAV
and the photos of the study area were obtained. The same input parameters were used in
all periods for the flights and an average of 190 photos were taken. Meteorological factors
were considered in shooting the aerial photos and the most suitable time periods were
chosen for the flights.
Table 4. Dates of flights

| Period | Flight Date | Flight Altitude (m) |
|--------|-------------|---------------------|
| 1 | February 17, 2016 | 100 |
| 2 | March 22, 2016 | 100 |
| 3 | April 9, 2016 | 100 |
| 4 | June 10, 2016 | 100 |
| 5 | July 21, 2016 | 100 |


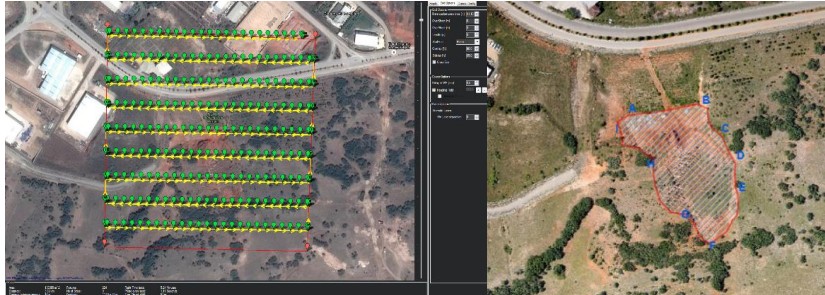




Figure 7a. Flight plan for the study area      Figure 7b. Borders of the landslide area
**_3.4. Point Cloud, 3D Model and Orthomosaic Production_**
The photos obtained from each flight period were stored in a computer with an
empty storage space of 100 GB and 8 GB of RAM. The photos were analyzed by using
the Pix4D software.
In the first stage, quality checks were performed for the images, dataset, camera
optimization and GCPs and these were calculated and the software produced the quality
check report for each of the time periods. The Ground Sampling Distance (GSD) is the
distance between two consecutive pixel centers measured on the ground. The bigger the
value of the image GSD, the lower the spatial resolution of the image and the less visible
details; GCPs are used to correct the geographical location of a project.
At least three GCPs are required to produce point cloud, orthomosaics and 3D
models, which come from the desired datum from the photographs taken. Optimal
accuracy is usually obtained with 5 - 10 GCP [22]. GCPs should also be well distributed
over the data area. To orient and balance the point cloud and the 3D model, Helmert
Transformation was applied. the transformation process was carried out with seven
parameters, which were generated from a minimum of three GCPs and point cloud
relations (Niethammer et al., 2011; Watson, 2006; Crosilla and Alberto, 2002).
In this study, the geographical location of the project was oriented and balanced through
the use of eight GCPs. The RMS and GSD values of GCPs are given in Table 5.

Table 5. GCPs' mean RMS errors

| Periods | RMS (mm) | GSD (cm/in) |
|---------|----------|-------------|
| #1 | ±23 | 3.11 / 1.22 |
| #2 | ±29 | 3.04 / 1.20 |
| #3 | ±28 | 3.50 / 1.38 |
| #4 | ±33 | 3.27 / 1.28 |
| #5 | ±18 | 3.57 / 1.40 |

The second stage increased the density of 3D points of the 3D model, which were
computed in the first stage. It represents the minimum number of valid re-projections of
this 3D point to the images. Each 3D point must be projected correctly in at least two
images. This option can be recommended for small projects, but it creates a point cloud
with more noise. The minimum number of matches is three in Pix4D, as a default, but up
to six can be chosen. This option reduces noise and improves the quality of the point
cloud, but it can calculate fewer 3D points in the endpoint cloud.
In this project, the number of matches was taken as three. The second stage results
are given in Table 6.
Table 6. Average density per m$^3$

| Periods | Average Density (per m$^3$) | Grid DSM (cm) |
|---------|------------------------------|---------------|
| #1 | 106.31 | 100 |
| #2 | 104.15 | 100 |
| #3 | 100.72 | 100 |
| #4 | 128.15 | 100 |
| #5 | 117.17 | 100 |



In the third stage, a Digital Surface Model (DSM) and an orthomosaic were
formed for all periods. DSM formation was achieved by the triangulation method with
100 cm grid intervals. The aspect maps, showing the landslide motion direction for the
first and last periods, were derived by using the DSMs of periods 1 and 5. The differences
between these maps can be seen, especially in the western and northern areas (Figure 8).
This means that there was a movement between periods.

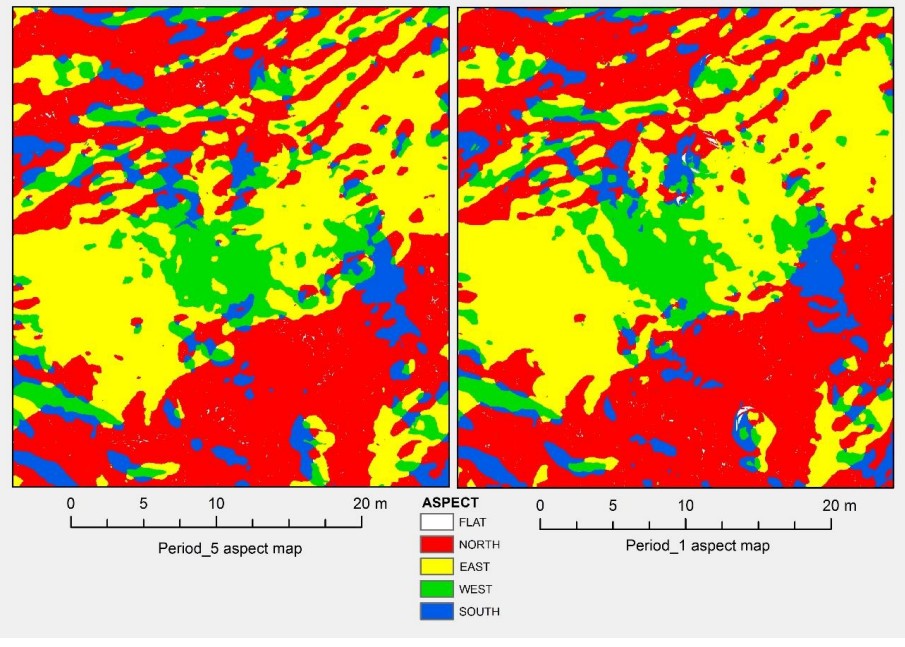

Figure 8. Aspect maps of period 5 (left) and 1 (right).

*3.5. Analysis of the Point Clouds, 3D Models and Orthomosaics*
Seventy-three object points were determined in the study area in order to monitor the
speed and direction of the landslide movement (Figure 9). These points, which represent
the topography, were chosen from the clearly visible details in the model and the field.





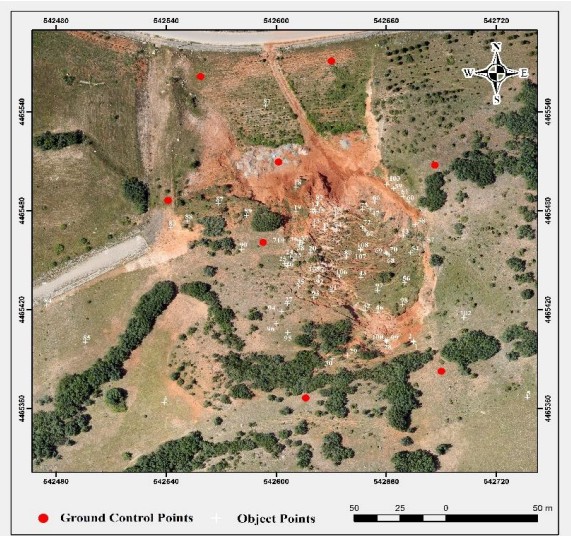


Figure 9. Ground Control and Object points

The 3D position information, orthomosaics and DSMs of the object points were produced
in each period. The 3D position data were compared consecutively. As a result of these
comparisons, differential displacements were calculated between T2 and T1, T3 and T2,
T4 and T3, T5 and T4, and are given in Figures 10, 11, 12 and 13. Additionally, Figure
14 provides a diagram showing the two-dimensional position shift (Δs) and height (ΔH)
changes between T5 and T1 (the last and the first periods).
According to these diagrams and Table 7:
a)  Points shown with a star (*) are at the centre of the area of motion and their
positional displacement is higher than the median value (>21 cm),
b)  Points shown without a star are outside the landslide area and their positional and
height displacement values are lower than the median value (<21 cm).


Figure 10. T2-T1 period differences





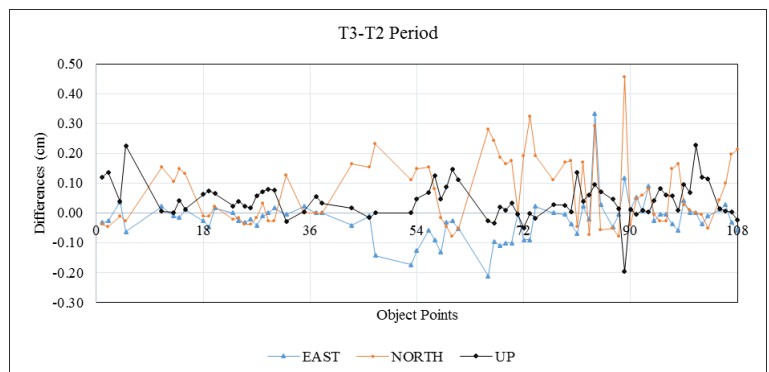


Figure 11. T3-T2 period differences


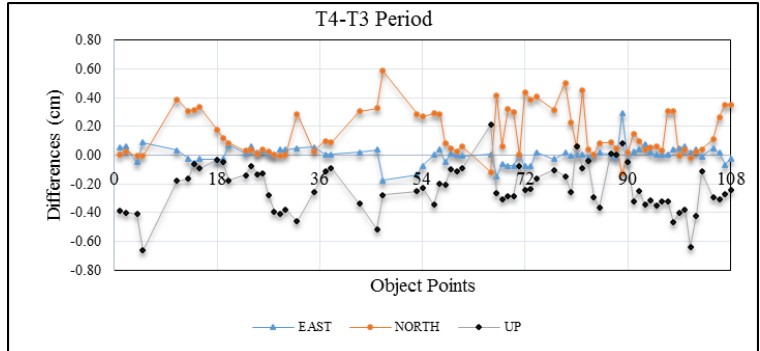


Figure 12. T4-T3 period differences


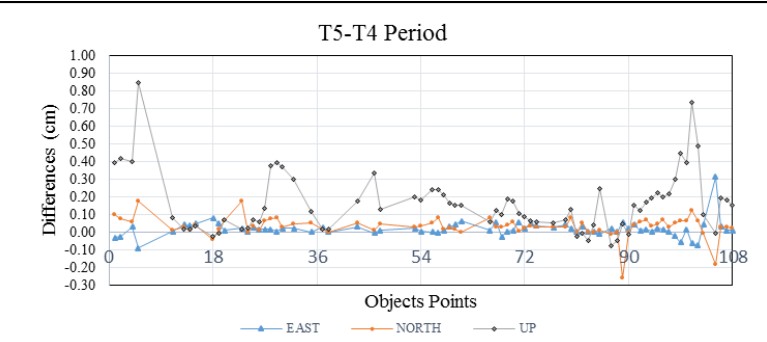


Figure 13. T5-T4 period differences





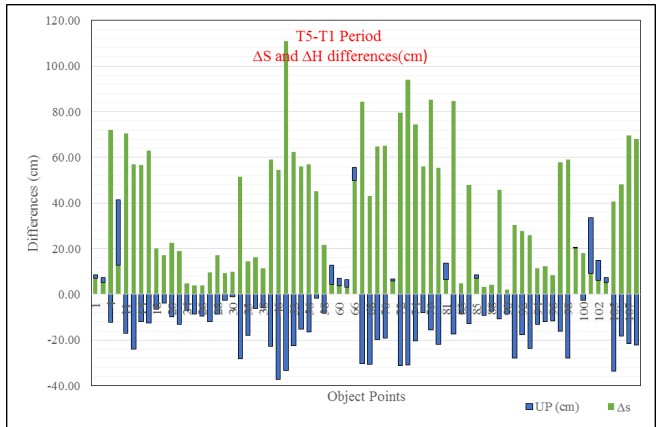


Figure 14. T5-T4 period ΔS and ΔH differences (cm)

The maps in Figure 8 show that the points with high positional displacement also had
a change of height by 70%. The positional and height displacement correlation coefficient
was calculated as *σ=0.73*. Thus, position and height changes are highly related to each
other.

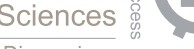
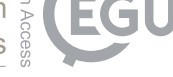
Table 7. Vertical and horizontal motion magnitudes (cm) of the points

| Bigger than median movement value (>21 cm) | | | Smaller than median movement value (<21 cm) | | |
|---|---|---|---|---|---|
| Number of Object Points | Movement of Δs (cm) | Movement UP (cm) | Number of Object Points | Movement of Δs (cm) | Movement UP (cm) |
| 47* | 111.0 | 33.2 | 18 | 20.3 | 6.4 |
| 73* | 94.0 | 31.0 | 23 | 19.0 | 13.3 |
| 79* | 85.3 | 15.5 | 100 | 18.0 | 2.5 |
| 82* | 84.8 | 17.4 | 28 | 17.2 | 8.6 |
| 67* | 84.4 | 30.2 | 19 | 17.1 | 3.8 |
| 72* | 79.7 | 31.2 | 37 | 16.4 | 6.3 |
| 74* | 74.6 | 20.3 | 35 | 14.5 | 17.8 |
| 4* | 72.1 | 12.2 | 5 | 12.9 | 5.0 |
| 11* | 70.6 | 17.1 | 95 | 12.2 | 12.0 |
| 107* | 69.7 | 21.7 | 94 | 11.4 | 13.2 |
| 108* | 68.2 | 22.1 | 38 | 11.4 | 5.8 |
| 70* | 65.1 | 19.2 | 30 | 9.9 | 1.1 |
| 69* | 64.8 | 19.6 | 27 | 9.8 | 12.0 |
| 15* | 63.0 | 12.5 | 29 | 9.5 | 2.5 |
| 53* | 62.4 | 22.4 | 101 | 9.1 | 5.0 |
| 43* | 59.1 | 22.9 | 96 | 8.5 | 11.6 |
| 98* | 58.9 | 27.8 | 77 | 8.0 | 8.0 |
| 97* | 57.8 | 16.1 | 85 | 7.2 | 1.5 |
| 13* | 57.0 | 24.0 | 1 | 7.0 | 1.6 |
| 56* | 56.8 | 16.6 | 81 | 6.4 | 7.2 |
| 14* | 56.7 | 11.9 | 102 | 6.2 | 8.7 |
| 54* | 56.1 | 15.3 | 71 | 5.8 | 1.0 |
| 80* | 55.6 | 21.9 | 103 | 5.4 | 1.8 |
| 46* | 54.7 | 37.3 | 2 | 5.3 | 2.0 |
| 32 | 51.6 | 28.2 | 66 | 5.0 | 5.6 |
| 106 | 48.3 | 18.1 | 83 | 4.9 | 8.7 |
| 84 | 47.9 | 13.0 | 24 | 4.8 | 7.2 |
| 89 | 45.7 | 10.6 | 59 | 4.3 | 8.4 |
| 57 | 45.3 | 1.7 | 88 | 4.1 | 7.5 |
| 68 | 43.0 | 30.7 | 26 | 4.0 | 9.4 |
| 105 | 40.8 | 33.8 | 25 | 3.8 | 8.6 |
| 91 | 30.3 | 27.8 | 60 | 3.7 | 3.4 |
| 93* | 26.0 | 23.6 | 87 | 3.4 | 9.2 |
| 20* | 22.6 | 9.7 | 61 | 3.2 | 3.1 |


As a result of the positional movements obtained in the landslide area, point velocity
vectors (*Vx, Vy, Vz*) were calculated using Equation 2 below, and they are given in Table
8. It was found that the general characteristic surface movement of the landslide took
place in the north-south direction (Figure 15).

$$V\{x, y, z\} = \frac{\Delta V\{x,y,z\}}{\Delta t} * 365 \qquad (2)$$
Here:
*Δt:* T5-T1 periods time difference,
*ΔV {x,y,z}:* The difference between Cartesian coordinate components between the T5 and
T1 periods.





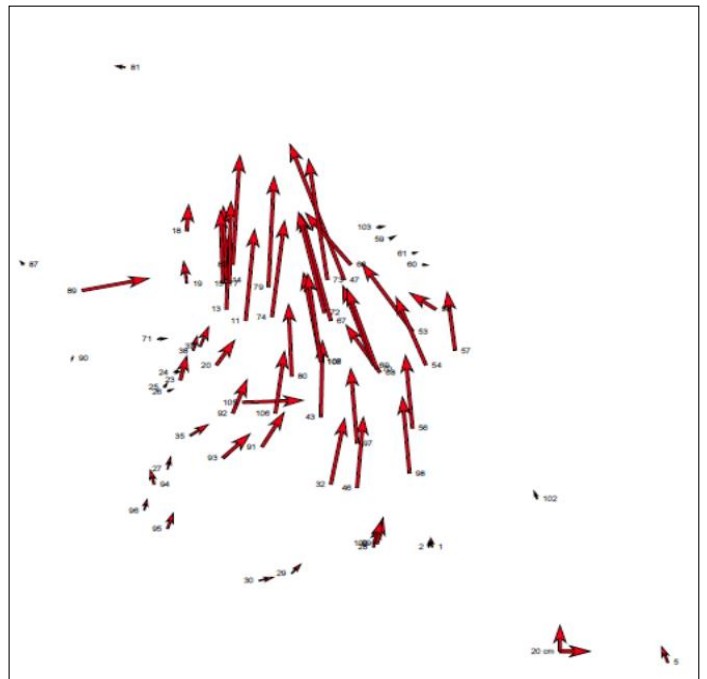


Figure 15. Characteristic surface movement of the landslide (m/year)

According to the velocity vectors, it may be seen that the landslide did not display
a typical structure.  The maximum movement was found to be $v_x = -2.095$ m, $v_z = -2.932$
m and $v_z = 2.036$ m.
Table 7 and Figure 14 show that the object points numbered #47, 73, 79, 82, 67,
72, 74, 4, 11, 107, 108, 69, 70, which were at the centre of the movement and had
positional (2D) displacement (>50 cm). The object points numbered #29, 101, 77, 96, 01,
85, 71, 81, 102, 02, were outside the center of the movement and had positional (2D)
displacement (<10 cm).



Table 8. Object points annual velocity vectors

| #Object No | Vx (m/year) | Vy (m/year) | Vz (m/year) | #Object No | Vx (m/year) | Vy (m/year) | Vz (m/year) |
|---|---|---|---|---|---|---|---|
| 1 | -0.068 | -0.095 | 0.219 | 68 | -0.851 | -1.605 | 0.279 |
| 2 | -0.064 | -0.023 | 0.186 | 69 | -1.111 | -1.700 | 1.189 |
| 4 | -1.214 | -1.568 | 1.593 | 70 | -1.122 | -1.685 | 1.212 |
| 5 | 0.474 | 0.171 | 0.966 | 71 | -0.108 | 0.172 | 0.036 |
| 11 | -1.767 | -1.035 | 1.480 | 72 | -1.721 | -2.010 | 1.362 |
| 13 | -1.583 | -1.084 | 0.968 | 73 | -2.095 | -2.077 | 1.772 |
| 14 | -1.241 | -0.996 | 1.233 | 74 | -1.955 | -1.063 | 1.505 |
| 15 | -1.435 | -1.001 | 1.387 | 77 | -1.159 | -0.913 | 1.306 |
| 18 | -0.530 | -0.333 | 0.392 | 79 | -1.958 | -1.268 | 1.908 |
| 19 | -0.346 | -0.343 | 0.364 | 80 | -1.434 | -1.139 | 0.981 |
| 20 | -0.804 | -0.064 | 0.285 | 81 | 0.265 | -0.079 | 0.191 |
| 23 | -0.707 | -0.335 | 0.192 | 82 | -2.009 | -1.260 | 1.853 |
| 24 | -0.261 | 0.013 | -0.148 | 83 | -0.052 | -0.177 | -0.293 |
| 25 | -0.284 | -0.118 | -0.109 | 84 | -1.588 | 0.275 | 0.615 |
| 26 | -0.306 | -0.066 | -0.171 | 85 | -0.147 | 0.016 | 0.206 |
| 27 | -0.472 | -0.255 | -0.017 | 87 | -0.200 | -0.239 | -0.136 |
| 28 | -0.575 | -0.234 | 0.246 | 88 | -0.048 | -0.151 | -0.253 |
| 29 | -0.311 | 0.037 | 0.133 | 89 | -1.317 | 0.964 | 0.001 |
| 30 | -0.268 | 0.214 | 0.043 | 90 | -0.136 | -0.124 | -0.252 |
| 32 | -1.716 | -0.857 | 0.711 | 91 | -1.379 | -0.373 | 0.073 |
| 35 | -0.776 | -0.059 | -0.181 | 92 | -1.044 | -0.355 | 0.289 |
| 37 | -0.534 | -0.140 | 0.263 | 93 | -1.216 | -0.108 | -0.040 |
| 38 | -0.380 | -0.164 | 0.164 | 94 | -0.429 | -0.430 | -0.002 |
| 43 | -1.585 | -1.112 | 1.050 | 95 | -0.544 | -0.240 | 0.037 |
| 46 | -1.874 | -1.190 | 0.605 | 96 | -0.436 | -0.238 | -0.041 |
| 47 | -1.863 | -2.932 | 2.036 | 97 | -1.307 | -1.136 | 1.166 |
| 53 | -0.734 | -1.995 | 0.890 | 98 | -1.564 | -1.349 | 0.932 |
| 54 | -0.865 | -1.497 | 1.048 | 99 | -0.437 | -0.140 | 0.537 |
| 56 | -1.285 | -1.143 | 1.129 | 100 | -0.479 | -0.112 | 0.397 |
| 57 | -0.747 | -0.770 | 1.154 | 101 | 0.412 | 0.206 | 0.786 |
| 58 | -0.051 | -0.790 | 0.150 | 102 | 0.122 | 0.000 | 0.350 |
| 59 | 0.064 | 0.208 | 0.244 | 103 | -0.089 | 0.163 | 0.069 |
| 60 | 0.007 | 0.165 | 0.063 | 105 | -1.587 | 0.589 | -0.723 |
| 61 | -0.014 | 0.123 | 0.095 | 106 | -1.385 | -0.747 | 0.862 |
| 66 | 0.018 | -1.281 | 1.183 | 107 | -1.472 | -1.579 | 1.336 |
| 67 | -1.722 | -2.124 | 1.498 | 108 | -1.519 | -1.493 | 1.297 |

**4. Results and Conclusions**
As a result of this study, we found that unmanned aerial vehicles have undeniable
advantages in disaster management and they have clear benefits over other methods. The
monitoring process must be continued for taking necessary precautions in case of




continuity and acceleration of landslides. Monitoring the landslide velocity is not possible
with conventional systems. Firstly, it is not possible to monitor an ongoing movement in
areas where the ground movement is active using ground surveying methods. These
movements have to be monitored by using remote measurements (remote sensing,
photogrammetry and UAV). Aerial photogrammetry and remote sensing techniques are
not usually preferred as they are expensive, measurements cannot be made at the desired
time, and they cannot achieve the sensitivity obtained with UAVs.
This study was carried out with the aim of monitoring the landslide acceleration of
movement of an area that could lead to great danger if it continues. In this study, GSD
values of 3.11/1.22-3.57/1.40 cm/in were reached with a flight altitude of 100 m. It is not
possible to reach these values with manned aerial vehicles or satellite images because
flight altitudes will be higher in both cases and the result of this situation will decrease
the sensitivity. Thus, it was concluded that the most effective situational awareness and
monitoring might be achieved by UAVs. Additionally, if it is desired to increase
sensitivity in monitoring landslides, GCPs should be assigned in a suitable distribution
with a suitable geometry at places that are not affected by the landslide, and the area of
flight should be widened based on these GCPs.
This study shows that UAVs are important tools in determining the speeds and directions
of landslide movements. In addition, landslide movements may be monitored in real time
using UAVs, allowing decisions to be made and precautions to be taken. In the light of
the UAV data obtained, early warning may prevent more tragic disasters and the
necessary precautions can be taken. Another important issue that needs to be emphasized
at the end of this study is that, with other traditional methods, the monitoring of landslides
and determination of the speed and direction of movement in real time is impossible.

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
