# Peer review of "The Role of Unmanned Aerial Vehicles (UAVs) In Monitoring Rapidly Occuring Landslides"

_Natural Hazards and Earth System Sciences, 2018_

## Referee Comment (RC1) · Anonymous Referee #1 · 14 Feb 2018

This paper presents the use of a UAV for multi-temporal analysis of an active landslide. The degree of novelty is insufficient, and the paper seems to be a mere description of a well-known UAV application. Authors should emphasize the scientific novelty of their work to be considered for an important Journal as Natural Hazards and Earth System Sciences. In the following some suggestion to improve the work. A native English speaker should revise the language of the manuscript Line 26, the term subsidence for landslide activity is not correct. It is better to describe the change as vertical displacement. Line 33 sentence not clear Line38, this is not true. Authors seem to consider only shallow landslides. Why not rock falls. The considered bibliography in the introduction is very limited. Line 42-44: again, this is a simplified and not correct description of possible landslides triggers. From line 76: from the description of the presented work

in the introduction, the paper seems to be focused on a simple exercise for the use of a UAV for a multi-temporal acquisition of images on an active landslide. Authors should emphasize the scientific novelty of presented manuscript. Chapter 2 the presented system is similar to many commercial systems. just to cite an example, phantom 4 has more or less the same performance and can be bought online. Table 3: it is not clear where these points are. Figure 5: not necessary Figure 7 not necessary Chapter 3.4 this is the simple description of the PIX4D procedure, that is typical of every structure from motion software. Figure 8: the aspect map is not necessary. The publication of this map should be motivated by authors. The use of aspect map for the identification of movement is quite critical. Authors should be very careful in the use of this approach for finding movements. Line 279: what is an "object point"? I suppose that authors used the same approach of Niethammer, et al, but they have to explain better. Figure from 10 to 13: horizontal displacement? Figure 14 is not clear, and not enough described in the text. Discussion: This paper presents a straightforward application of a multi-temporal analysis of an active landslide. Many papers described more complex approaches. To have an idea, Authors could consider the Special issue on RPAS and natural hazard published on NHESS and in particular the review paper. The scientific novelty degree of the paper is very poor. Authors cited at line 215 that the acquisitions have been made after rainfalls and snow melting events, but then this data is not considered. A correlation between rainfall events and the landslide activity could be a possible interesting add value.

---

## Referee Comment (RC2) · Anonymous Referee #2 · 1 Mar 2018

In this paper, the authors explore the use of UAVs for rapidly-deforming landslide monitoring. At the first time I saw the title, I was very much excited because I expected an innovative approach of 4D monitoring of a landslide from UAVs, but in fact, the work is not such one but with rather an ordinary approach of repeated measurements at a monthly scale. Also, this manuscript can be strengthened if some more additional analyses are provided, including pixel-based movement detection of the landslide not only with the 73 points. Otherwise, the work remains just as a technical report and is unsuitable as a scientific paper for NHESS.

Below are some minor comments regarding the manuscript.

* Abstract. Please provide the information on the time periods of measurements (monthly).

* Introduction. The literature cited regarding the UAVs are relatively old. More recent, plenty amount of papers can also be cited. The last paragraph of Introduction should be presented in the Method or Study site section. Instead, please provide the research motivation – why UAVs for landslide monitoring, at what scale or frequency??

* Figure 3. This figure may be unnecessary. Also, please check there is no infringement of the copyright.

* Study area. Please provide more information on the landslide itself. When did it begin to slide? What triggered the landslide? Such basic information is missing. Furthermore, this long section includes the methods and results. Please reorganize.

* Figure 6. How was this image derived? Please suggest appropriate courtesy.

* Figure 8. Why is the aspect distribution presented? To show the differences, any other maps (hillshade, slope angle) should be more helpful.

* Figures 10-14. These figures can be merged into one. Moreover, showing the number data in the X-axis is rather meaningless.

* Line 366. What is the "typical structure"?

* Results and Conclusions. This section does not include Results, but some concluding remarks. The Discussion is completely missing. . .

* Line 420. I did not understand why the authors could say it "impossible" to monitor the landslide motion.

* Figures. In many figures, the scale and north-direction mark are missing. Captions are too short and not fully informative.
* * *

---

## Referee Comment (RC3) · Anonymous Referee #3 · 19 Mar 2018

The paper presents a typical application of a multi-epoch UAV-monitoring. Other reviewers have main concerns about the novelty of the methodology. I agree that the technology/methodology is not new but I also think the paper itself is rather focused on a UAV-based monitoring of a rapid-developing landslide rather than technological/methodological improvement of UAV based monitoring.

Rapidly-developing landslides require a high temporal resolution than the conventional terrestrial methods and UAV-based monitoring presents a feasible alternative. While UAV-based applications are not new, locations with well-known landslide history and rapid-development characteristics are rare. The application was carried out in Turkey where only few locations are feasible for this type of monitoring.

I believe the authors did a good job of revising the original manuscript. The authors

apparently considered the concerns raised by the reviewers and the current version seems to much more improved and mature to me.

---

## Author Comment (AC1) · 19 Mar 2018

A native English speaker was revised the manuscript. The term 'breakdown' was used instead of 'subsidence'. 1. Line 33 was edited as "Soil drifts are caused by two main factors, human and environmental effects in general. Human factors can be controlled; however, it is very difficult to control factors originating from topography and soil structure (Turner et al., 2015)." 2. Line 38 was edited as "The main reasons for the increase in landslide disasters are that they become more susceptible to instability of surface land because of extreme destruction of natural resources, deforestation, increased urbanization and uncontrolled land use." Additional statements can be found in the revised paper. 3. Additional statements can be found in the revised paper. 4. Additional statements can be found in the revised paper. The literature cited regarding the UAVs

are relatively old. More recent, plenty amount of papers can also be cited. 5. Line 42-44 was revised as "The main reasons for the increase in landslide disasters are that they become more susceptible to instability of surface land because of extreme destruction of natural resources, deforestation, increased urbanization and uncontrolled land use. Triggering can occur faster because of short or long periods of heavy rain, earthquakes, or subterranean activity (Lucier et al., 2014)." 6. Cxx 7. The multicopter was used for this study was designed and produced by the department of Geomatics Engineering at Gaziosmanpaşa University (GOP) and Teknomer Company in Techno park. Teknomer is one of the most important UAV producer companies of Turkey. Teknomer brand UAVs can be seen at http://akteknomer.com/ net address. This UAV was not produced only for this study. TEKONOMER GEO V2 multicopters have been producing for photogrammetric observation companies. 8. These poits coordinates were added the Table 3. 9. Figure 5 was deleted. 10. Figure 7 was deleted. 11. 000 12. The soil motion change in the landslide area hadn't been seen exactly in the aspect map. Because of this, it was deleted. Ἄȓnstead of aspect map excavation and repleacement of the earth material was investigated with DoD map which generated with integrated of first and last DSMs. In addition, in order to obtain landslide deformations, the DoD was applied by subtracting the first UAV-DSM from the last UAV DSM. Also seventy three sample poins 3D displacements was investigated with DoD map. It has been seen that the displacement direction and displacement values of sample points are compatible with two methods. 13. Object poits are sample points which was sellected on the study area. 14. Sample points location have changed between first and last observation periods. This displacement is three-dimensional (Y,X and H). The displacement of the sample point in the horizontal position (slip) was named horizontal displacement. 15. Additional statements can be found in the revised paper. 16. Total periodic precipitation amounts was submitted for observation dates. A correlation between precipitation events and landslide activity has not been studied.

2018-13, 2018.

---

## Author Comment (AC2) · 19 Mar 2018

1 Observations were carried out between 17.02.2016 and 21.07.2016, one measure per month. 2 The last paragraph presented in 2Study Area section. Resources have been updated by accessing new resources related to UAVs. 3 Figure 3 has been removed from the article. 4 Aaaa. 5 This image was derived from processed UAV images. 6 Aspect map was deleted from manusicript. DSM of difference (DoD) between the first and the fifth flight data was generated. Elevation difference map was generated using first and last flight. Seventy three sample points elevation differences between firs and last epoch was compared with produced DoD map. 7 Sample poins 3D displacent was caltulated between epochs and between first and last epoch. X and Y directions represent the North and East displacepent directions values of sample

points respectively. From X and Y displacement values horizontal displacements of sample points was calculated. $\Delta S=$ square ($\Delta X2+ \Delta Y2$). 8 The aggregate stability of the soil tests was under 46.2% and showed low aggregate stability with a high risk of soil movement. 9 Results and Conclusions sections coplatelly edited. 10 This section edited. Additional statements can be found in the revised paper 11 The scale and north-direction mark was added to all maps. Captions were revised.

---

## Author Comment (AC3) · 19 Mar 2018

This is to certify that the following document has been proofread in terms of the quality of the English used:

The Role of Unmanned Aerial Vehicles (UAVs)
In Monitoring Rapidly Occurring Landslides

The document was submitted for proofreading by Omer Yildirim.
.

The proofreader was Dr Hazel Bird on behalf of Academic Proofreading.

If you have any questions, please do not hesitate to ask. We can be contacted at prime@academicproofreading.com

Academic Proofreading
Hurworth House
17 Beechwood Terrace
Sunderland
SR2 7LY
United Kingdom

Dr. David Mercer
16th March, 2018

---

## Author Comment (AC6) · 19 Mar 2018

The comment was uploaded in the form of a supplement:
https://www.nat-hazards-earth-syst-sci-discuss.net/nhess-2018-13/nhess-2018-13-AC6-supplement.pdf

---

## Author Comment (AC7) · 21 Mar 2018

- You can see in attachment our revised paper. Referee 1 and Referee 2 comments were considered

Please also note the supplement to this comment:
https://www.nat-hazards-earth-syst-sci-discuss.net/nhess-2018-13/nhess-2018-13-AC7-supplement.pdf